# Analyzing Greece 2010 Memorandum's Impact on Macroeconomic and Financial Figures through FCM

**Stavros P. Migkos** [1,*] **, Damianos P. Sakas** [2] **, Nikolaos T. Giannakopoulos** [2,*] **, Georgios Konteos** [3] **and Anastasia Metsiou** [1]

1    Department of International & European Economic Studies, School of Economic Sciences, University of Western Macedonia, 501 00 Kozani, Greece; diees00004@uowm.gr
2    Department of Agribusiness and Supply Chain Management, School of Applied Economics and Social Sciences, Agricultural University of Athens, 118 55 Athens, Greece; d.sakas@aua.gr
3    Department of Business Administration, School of Economic Sciences, University of Western Macedonia, 511 00 Grevena, Greece; gkonteos@uowm.gr
*    Correspondence: diees00015@uowm.gr (S.P.M.); n.giannakopoulos@aua.gr (N.T.G.); Tel.: +30-6973822068 (S.P.M.); +30-6940013673 (N.T.G.)

**Abstract:** The financial crisis of 2008 has caused a series of drawbacks to economies around the world. Greek economy has been hit twice at 2009, since its credibility worsened, provoking the implication of harsh fiscal measures from the 2010 Memorandum of Understanding (MoU). The effects of these measures to Greek macroeconomic figures have been widely criticized. Authors aim to estimate these effects at the macroeconomic figures of Greece through utilization of Decision Support Systems, and propose accurate insights regarding their efficacy. By capitalizing on regression analysis and Fuzzy Cognitive Mapping processes, specific results from 2010 Memorandum's measures arise. It has been calculated that measures implied by 2010 Memorandum have been harsh and posed a negative effect on key Greek macroeconomic figures like GDPR, public debt, etc., especially with the ongoing 2008 financial crisis.

**Keywords:** 2010 memorandum; macroeconomics; financial crisis; monetary policy; regression analysis; fuzzy cognitive mapping; decision support systems

## 1. Introduction

Ever since Eurozone's (EZ) inception, participating countries have begun to develop asymmetrical deficits (Clifton et al. 2018). The so-called "public sector chaos" is among the primary reasons for the EZ crisis (Galenianos 2015). As a result, the three most often tracked macroeconomic deficits are the Government Budget deficit, the amount of public debt, and the deficit in the equilibrium of current transactions, all expressed as a percentage of GDP. Therefore, this is in opposition to the nations of Central and Northern Europe, whose financial conduct has not been the cause of the periphery nations' crises (Galenianos 2015). Exterior inefficiencies, as was proven in the years after the recession, were the true root of the issues.

With regards to a nation's global economic status, it is calculated by subtracting the overall external capital held by individuals residing in the country from the total internal investment possessed by abroad residents (Swiston 2005). Accordingly, whenever there is a deficit in the current transactions' equilibrium, a nation's investment position drops, whereas it grows if there is an equilibrium surplus (Galenianos 2015). Before joining the Euro, no nation used to have a substantial current transactions equilibrium deficit (e.g., Portugal's deficit just was 2.5 percent greater than GDP), while peripheral nations displayed a major worsening after the adoption of the euro. As a result, it can be discerned that the periphery nations, after joining the EZ, have gained fairly negative foreign investment situations, and net external indebtedness reached 60% of GDP.

The adoption of the euro enhanced financial unity by removing currency fluctuation rates, resulting in the uniformity of financial policies across the Eurozone (Kalemli-Ozcan et al. 2010). Such an achievement was interpreted as an indication of the euro's accomplishment, with economic growth being amongst the currency's fundamental aims. Deficits of peripheral countries' current transactions equilibrium indicate capital transfer all across the EZ.

Economic crisis originally impacted banking firms, with the EZ bearing the brunt of the damage, without discounting the accountability among its participating nations and the organizations themselves regarding the length and severity of the crisis (Wolf 2013). As soon as May 2010, the Greek recession began to extend to the remainder of EZ, as seen by the expansion of the EZ Credit Default Swap (CDS), lower bond rates, and a decrease in share exchanges (BIS 2010).

Upon Greece's creditworthiness reduction after 2009, Greek bond interest prices skyrocketed, prompting the administration to announce austerity and deficit reduction policies (Lim et al. 2018). A mistrust of such actions, combined with the heavy price of funding the Greek economy, resulted in a growth in state debt. With the increase in the borrowing rate, the debt grew to 7.8 percent in April 2010 over 4.6 percent in October 2009, with Greece unable to fulfill the funding requirements of 60.8 billion to fund the debt's closure expenses, and also wages and pensions.

The purpose of a unified monetary policy is to stabilize the economies of its nations to safeguard them from instability and bankruptcy in the event of a catastrophic crisis. Via arrangements among participating nations, the European Union implements shared monetary and currency rates management regulations and tactics targeted at addressing any economic challenges that could occur inside the EZ (Baldwin et al. 2006). EZ nations present a convergence of three years and concrete fiscal policy targets are specified, depending on the requirement of the euro and every nation, following the directives of the European Union's executive committee, the entity accountable for assessing such programs on their own (Porte et al. 2001).

This approach results in the supply of ideal conditions in EU arrangements with non-euro nations, the integrity of its participating nations' commerce, the competitiveness of the currency, etc. (Cini and Borragán 2016). Throughout our research, the authors aimed to render valuable insights regarding the estimation of 2010 Greece's Memorandum measures effect to key figures of the Greek economy. By doing so, the tools for analyzing various effects and interactions among economic factors and variables broaden. Hence, the research's main purpose is to scrutinize the 2010 Greek Memorandum's impact on vital economic figures (e.g., GDPR, Public Debt, Equilibrium of Current Transactions, Government Budget, etc.) to evaluate the efficiency of such policies and their repercussions on European economies.

This paper is structured as follows: in the first phase, an introduction to the main topics of the research takes place. It is followed by an analytical literature review, elaborating and referring to chronological events before and after the outbreak of the financial crisis, as well as presenting the research hypotheses and the theory of utilizing an exploratory and diagnostic model for simulating the 2010 Memorandum's effect on various key economic figures of the Greek economy. Next, in phase three, linear regression and simulation models are produced and their results are analyzed in Section 4, where outcomes are presented thoroughly. In the last sections, the paper's conclusions and overview of its principal results and suggestions are conducted, with the final phase focusing on providing authors' views regarding the trajectory of future research on the specific field.

### 1.1. Analysis of Greece's Macroeconomic Imbalances and Causes of the Debt Crisis

Considering their lack of independence over currency depreciation, nations with competitive susceptibility and reliance on external finance will suffer an internal devaluing procedure, as Greece did (Rathgeb and Tassinari 2020). The extent of financial imbalances caused by excessive public debt, economic dependency of nations, macroeconomic differences inside the EU, limited fiscal flexibility, monetary integration tightness, and a loss

of economic regulatory instruments to counterbalance a future economic recession and reduction in size (Bieler et al. 2019; Costa et al. 2016), just like it occurred in Greece. In Figure 1 the course of Greek macroeconomic figures over the last 15 years are shown.

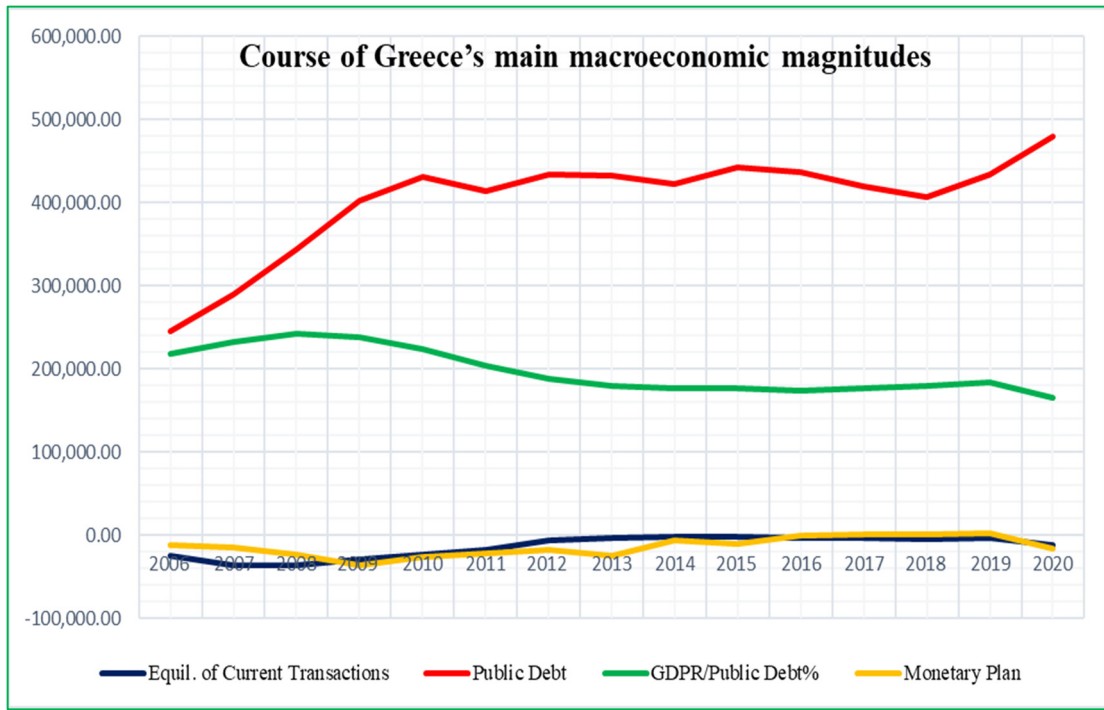

**Figure 1.** The course of Greece's main macroeconomic figures. Source of data: www.statistics.gr (accessed on 20 September 2021) and www.ec.europa.eueurostat (accessed on 20 September 2021).

Concerning the Greek economy's systemic issues, before the recession, foreign direct investments remained concentrated in moderate technological industries such as retailing and domestic services. Such conditions evolved as a result of various industries in Greece which are defined by oligopoly (Lall 1974). Other variables, such as the exchange rate's goodwill about the cost of goods and services, and the assumption that the proportion of foreign direct investment was directed at purchasing local firms, altered the nationality of property ownership instead of boosting the country's economic base. Apart from specialized enterprises, the non-diffusion of production methods in domestic manufacturing does not credit the accomplishment of competitive advantage in the industrial economies (Krugman 1991).

Pricing and salaries rose dramatically as a result of direct investment from the EZ's constituent nations, but this was not matched by an improvement in productivity, resulting in a steady loss of competitiveness (Galenianos 2015). Over the prosperous period, government performance in peripheral nations has diminished and malfeasance has soared (Fernandez-Villaverde et al. 2013). The easiness with which the private and public sectors would loan ultimately resulted in a much more complacent economic and political climate that inhibited changes. There had been insufficient stakeholder monitoring of National Governments' fiscal and macroeconomic progress, no efficient crisis management mechanisms, and thus no unified financial and monetary union.

Studies argue that the Greek economy lags well behind European South in terms of competitiveness and performance (Savelyev et al. 2019), with most economies, except Greece, succeeding in decreasing unemployment to European average levels. Additionally, it should be underlined that the Greek economy will continue to be defined by aspects other than a large public sector, a huge bureaucracy, poor institutional development, and heavy taxation of private enterprises (Siskos and Rogach 2014).

To accomplish fiscal reconfiguration, Greece must enhance expenses oversight as well as the extent of quality of government revenues, improve public debt management protocols, rearrange assets to attain more investments for economic growth, act swiftly to optimize the public sector, and identify non-tax earnings shortfalls.

### 1.2. Financial Support Mechanism and 2010 Memorandum's Fiscal Consolidation Measures

Because of the links connecting Europe's state financial institutions, there has been concern that what began as a public debt issue in Greece might escalate into a larger financial catastrophe (BIS 2010). As a result, European governments have taken activities to strengthen their financial institutions to avoid possible a larger bank panic as a consequence of a widespread bank capital reform using public money and individual deposits (Quaglia et al. 2009). Consequently, global commerce, investment, financial institutions' capacity to loan to people and enterprises, and consumer spending were all impacted.

European Monetary Union (EMU) national governments, and organizations like the ECB as well as the Eurogroup, seized on the role of handling the fiscal and financial crises of "turbulent" nations, making risky measures without advancing the concerns of the countries involved, despite their political objectives (Papastamkos and Kotios 2011). EZ's economic issues had an unequal effect on various EMU members (Revuelta 2021). Marketplaces eventually uncovered EMU's failure to handle the Eurozone's financial condition and react in a suitable way to reestablish its Member States' profitability.

This financing was disbursed in increments beginning on 18 May 2010 and will be finished as long as the Greek government follows the provisions of the Memorandum of Understanding (MoU). The surveillance and assessment of conformity with the requirements were conducted by the "troika", a triad of EC agencies comprised of the ECB and the IMF (European Commission 2010). The conditions of the MoU included budgetary changes totaling 14.5 percent of Greek GDP through 2014. It was intended to transform the main trade imbalance of 8.6 percent of GDP in 2009 into a primary surplus of 5.9 percent by 2014, as well as to reduce public debt beginning in 2013 when this would achieve a peak of 149.6 percent of GDP.

Cutbacks in government employee wages, a decline in the number of government employees, pension reform, a decrease in development and financing expenditure, a restructuring and decline in municipal authorities, a rise in valuation added tax (VAT) and alcohol, a special contribution to company earnings, enhanced property taxes, and the struggle against tax avoidance were among the primary fiscal initiatives examined. The fundamental financial initiatives primarily discussed address modernizing and reformation of the pension and healthcare programs, and also taxation, reinforcement of the banking system's solvency buffers, and restructuring of the financial regulatory regime.

There is little question that the Greek economy's bleak development expectations are the product of ineffective policies and organizations plagued by budgetary upheaval. Nevertheless, the actuality that the Greece–Portugal–Spain design, which has been heavily affected by the EZ recession, requires huge bailout funds and straightforward government bond investments by the ECB for equity capital must not prompt to the presumption that enforcing fiscal restraint inside the EZ is still the upper and foremost primary concern of policy-making (Katsimi and Moutos 2010), as chronological and financial evidence from these nations do not support this subject.

### 1.3. EU and ECB Fiscal and Monetary Policies' Effect on the Greek Economy

In 2009, the euro had established a context of minimal inflation and low-interest costs (including previous high-inflation nations) favorable to long-term prosperity, with clear indications of the currency's performance. However, many economists have already voiced doubts about the viability of a European unified monetary system (Friedman 2007). This view was founded on the underlying logic. First, economists argued that even if governments retained solid fiscal practices, the effect of asymmetrical shocks would indeed be mitigated (Krugman 2012). Second, they reasoned that, given less need

for demand measures to deal with external imbalances, national governments would implement fundamental changes. Third, economists reasoned that because the EZ would remove currency price risk off national interest rates, that would be simpler to identify credit risk and, as a result, investing possibilities across borders. To put it another way, removing international exchange volatility could result in stronger market regulation in government agencies (Fernandez-Villaverde et al. 2013).

Whereas external deficits were self-correcting, according to the gold standard, they became self-perpetuating within Eurozone due to the illusion of no systemic risk. The variance of the euro's rate of exchange is particularly challenging to predict. Nonetheless, EU and ECB's currency exchange policies have had an impact on both the rate of exchange and fluctuation. The ECB's measures, for instance, have contributed to reducing euro instability, implying that they should have helped to lessen financial instability and stabilize markets (Ehrmann et al. 2013). However, for the nations of Southern Europe, events have resulted in the instability of their economies and key macroeconomic figures.

The administrations of the member nations kept these set rates by being prepared to purchase or trade gold at such set rates on request (Eichengreen 1996). Income and pricing elasticity, on the other hand, is a required but just not adequate requirement for the functioning of a system of stable exchange rates. What is critical is indeed the presence of a structural adjustment program that causes the appropriate salary and pricing (Dellas and Tavlas 2013). As a result, there was no procedure for correcting economic and financial expansion, and Greece was able to manage significant current transactions and fiscal imbalances without implementing regulatory correctives (Dellas and Tavlas 2013).

Greece's current balance shortfall grew from 11.5 percent of GDP in 2001 (the same year the country entered the euro) to 18 percent in 2008. Extremely big and sustained foreign shortfalls are not anticipated under one well functioned and fixed currency rate regime. Because of the growth in Greek inflation, the actual interest cost fell, resulting in increased lending. Greater government lending has harmed the country's economy in two ways. Firstly, because Greek manufacturers deliver a diverse range of commercial items, the demand slope has a negative slope. Secondly, even as the government lent and spent more, non-tradable commodities costs rose compared to necessities. Earnings mostly in the non-tradable sector were raised in connection to salaries in the field of tradable goods.

At the request of international creditors, the Greek government imposed pension and tax restructuring, resulting in a decline in pensions and a considerable upsurge in taxes (Angeletos and Dellas 2013). There was even a deterioration of global competitiveness to the degree that pricing in certain sectors never could sustain the additional cost, for example, since costs never could rise owing to foreign competitiveness or the gain in production was inadequate to balance the increase in spending (Provopoulos 2014). The current budget deficit, on the other hand, resulted in the buildup (mostly) of state liabilities.

Initial funding was done via EZ monetary regulatory procedures, however owing to a shortage of suitable assets, the financial institutions progressively depended on the Bank of Greece's emergency liquidity assistance (ELA). Funding via ELA was more expensive than borrowing via monetary policy procedures. As a result, it began as a cash flow issue projected to become a solvency issue (Micossi 2015). International finance including retail financial markets was confiscated, primarily hurting financial and commodities markets in several periphery nations and intensifying the debt crisis. The financial intermediation for monetary policy almost nearly failed to work (Forbes et al. 2015). In Figure 2 below, the trajectory of the 4 main macroeconomic features of Greek economy can be seen, right before and after the implication of the MoU's measures.

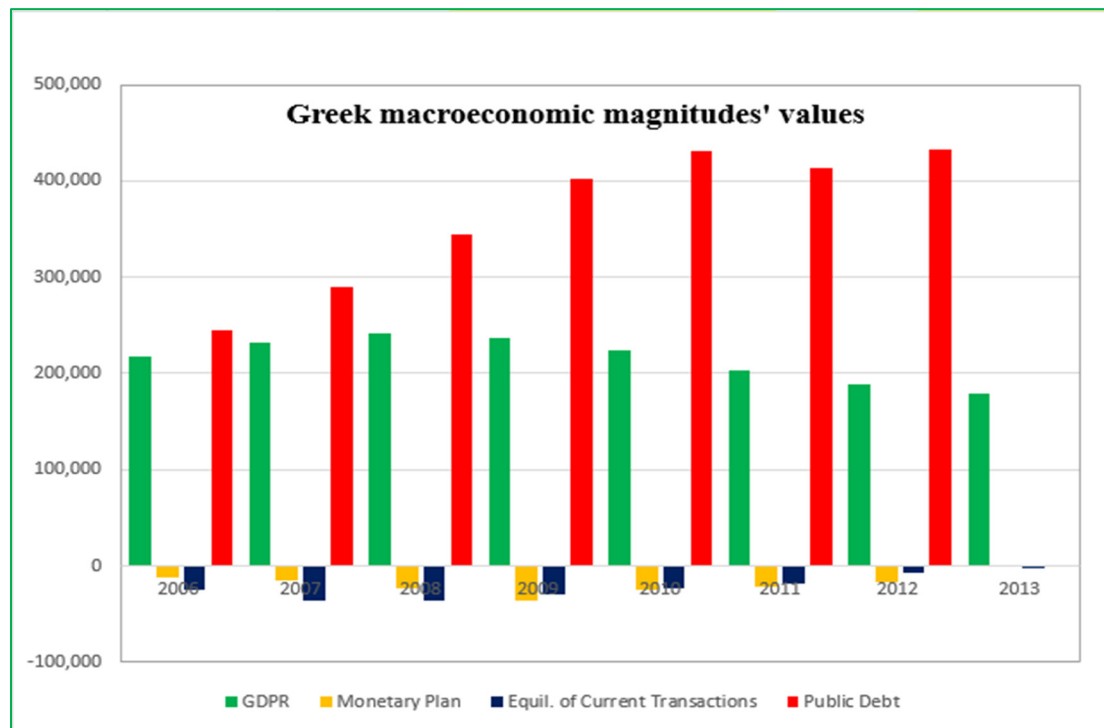

**Figure 2.** The trajectory of the four main macroeconomic figures of Greek economy before and after 2010 MoU's measures. Source of data: www.statistics.gr (accessed on 20 September 2021) and www.ec.europa.eueurostat (accessed on 20 September 2021).

### 1.4. Research Motivations

In order to obtain a better grasp of the paper's research focus and motives, a thorough elaboration of relative topics is presented. Placed in the center of the research is the assessment of the effectiveness of Greece 2010 MoU's measures based on their effect on Greek macroeconomic figures. Markantonatou (Markantonatou 2013) argues that the proposed measures of "internal devaluation" and the misunderstanding of labor costs and public expenditure led to harsh fiscal measures by performing socio-political analysis of their consequences. The whole healthcare sector, including pharmaceutical industries, were also heavily impacted during 2009–2013 (Simou and Koutsogeorgou 2014) as a result of measures imposed by the Government, following MoU's proposals. Through utilizing quasi-experimental analysis, Revuelta (Revuelta 2021) has stated that Greek GDPR has been significantly reduced during the period after the implication of the 3 Economic Adjustment Programmes, due to false perception of the needed measures for Greek economy. Zartaloudis (Zartaloudis 2013), on the other hand, highlighted that MoU's measures fired a series of political, social, and economic shifts that, as a result, did not achieve overall consensus on facing Greece's perpetual deficiencies. The present study is orientated to adding more scientific and simulation outcomes to the existing literature of 2010 MoU measures' impact on Greek economy by performing simulation scenarios and evaluating their effects on key macroeconomic features like GDPR, Public Debt, Equilibrium of Current Transactions, and Government Budget.

### 1.5. Approach of the Study

The aim of this study focuses on assessing the 2010 Memorandum's efficiency through evaluating its instant outcomes to Greece's financial performance. To do so, financial performance's key indicators should be defined. The authors reviewed related literature in the process of data curation and methodological approach determination. More specifically, the authors took into account the fact that previous literature reveals that some asset restrictions and macroeconomic policies can be successful in decreasing financial volatility, however

such interventions are typically incapable of reaching their declared goals (Beckmann and Czudaj 2017a). Currency deposits' significance is linked to public debt, as well as the influence on assumptions and macroeconomic stability (Beckmann and Czudaj 2017b). Variations in fiscal policy ambiguity could reflect a portion of the vast spectrum of fiscal factor predictions (Beckmann and Czudaj 2020). Beckmann and Czudaj's (2017c) approach is substantially extendable to other markets, allowing for the creation of a composite fiscal policy ambiguity indicator for the Eurozone. Thus, the authors performed statistical and exploratory analysis to key financial figures of Greek economy, such as GDP, etc., before and after the enactment of 2010 Memorandum's measures, to evaluate its effect through the figures' variation.

## 2. Materials and Methods

Aiming to fill the gap on the research mention in the previous section of motivations, the authors utilize statistical and simulation tools that fit the characteristics of the study. After collecting specific chronological values of Greek economy's macroeconomic figures, the authors started by performing correlation and linear regression analyses to the data. Prior to these analyses, data curation and validation test, in terms of data following the Normal distribution, were executed. The reasons supporting the selection of correlation and linear regression analyses are the aim of discerning variables' linear and causal relationships accordingly. Linear regression models are widely preferred over other methods in multiple occasions and sectors, frequently in cases of psychology (Gomila 2021), unemployment causes analysis (Abdulhamed et al. 2021), health studies (Kumari and Yadav 2018), etc. Most importantly, the reasons for linear regression method's utilization are based on the effectiveness of representing causal effects estimation and when there is no apparent ground supporting the use of more complex nonlinear methods (Gomila 2021). In addition, data were shown to develop high liner relationships, leading to the choice of simple linear regression models that can strongly strengthen the cause for their selection. This will support the verification or rejection of the research hypotheses and provide the necessary coefficients for the simulation analysis that follows.

Moreover, apart from the statistical analysis needed for verifying or rejecting study's hypotheses, a suitable simulation analysis through an exploratory model was performed. Fuzzy Cognitive Mapping (FCM) website platform application was selected as the simulation development tool for the study. A plethora of studies supporting the importance of FCM analysis has been published over the recent years. More specifically, applications of FCM can be seen in various sectors, especially in economic studies (Neocleous and Schizas 2012; Gini 2015), modeling investment (van Vliet et al. 2010), risk analysis assessment (Bakhtavar et al. 2021), and decision-making in environmental problems (Papageorgiou and Kontogianni 2012). At the next stages of the methodological framework, as referred above, and after setting the research's hypotheses, regression and correlation analyses were performed followed by the simulation deployment with the FCM analysis.

### 2.1. Research Hypotheses

At the research's current phase, the extraction of the paper's main research hypotheses takes place. Through this process, and based on the literature review's points, authors extract the research hypotheses that can support the provision of valid outcomes regarding the evaluation of the 2010 Memorandum's effect on Greek macroeconomic figures. The need to evaluate the impact of the 2010 Memorandum measures on the Greek economy is stated across the theoretical part of the paper. Such an assessment can be accomplished by defining the main macroeconomic figures of an economy, like GDPR, Public Debt, etc., and performing linear regressions and correlation analysis to identify a valid pattern of the Greek economy's features that cause significant variations to them. Through observing the relationship of the Greek economy's key 26 economic features (e.g., product taxes, inflation rate, net savings, abroad residents, etc.) with its main macroeconomic figures (GDPR, Public Debt, Equilibrium of Current Transactions, Government Budget) during

the 2010 Memorandum, the authors will be able to comprehend the overall significance of Memorandum's measures to Greece's economy. In a way, via this framework, the evaluation of the 2010 Memorandum measures' efficiency could be performed. Hence, the research hypotheses that back the referred methodological and research framework are mentioned below (Figure 3):

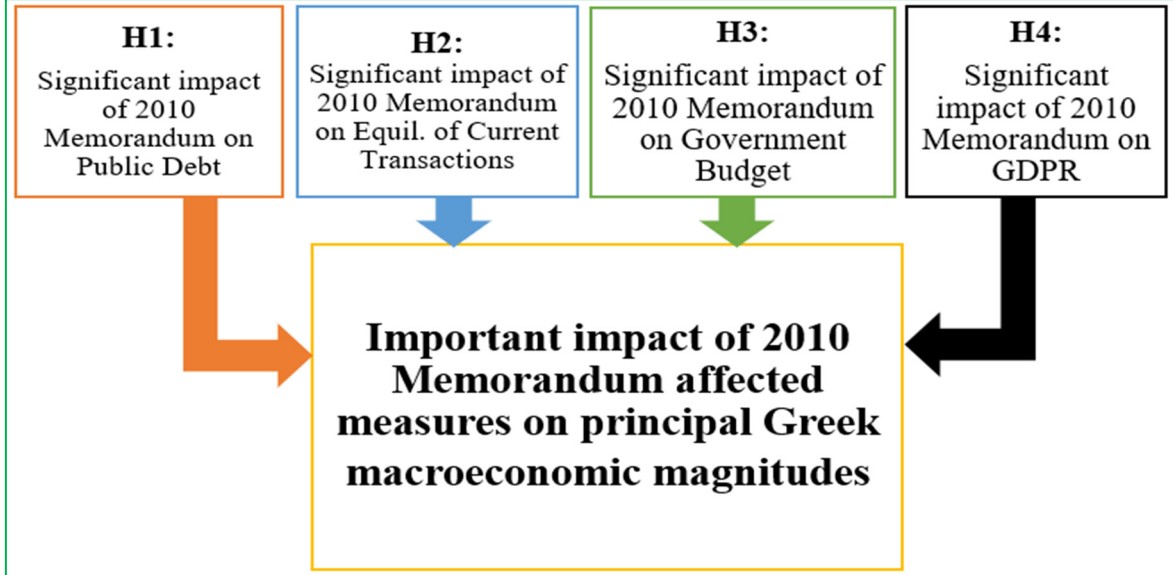

**Figure 3.** Conceptual framework.

**Hypothesis 1 (H1):** *The impact of the affected by the 2010 Memorandum Greek features to Public Debt is significant.*

**Hypothesis 2 (H2):** *Greece's Equilibrium of Current Transactions is being significantly affected by the variation of Greek economic features caused by the obtrusion of the 2010 Memorandum.*

**Hypothesis 3 (H3):** *A significant effect is provoked to Greece's Government Budget from Greek economic features afflicted by Memorandum of 2010.*

**Hypothesis 4 (H4):** *Key economic features, affected by the Memorandum of 2010, significantly impact Greece's GDPR figure.*

*2.2. Sample Selection and Data Retrieval*

For this level of analysis, the authors make effective use of the data collected from the sample collected from the websites of ELSTAT (ELSTAT 2022) and EUROSTAT (2022) and refer to the years 2000 up to 2020, aiming at the development of an explanatory model that will focus on the statistically significant positive and negative interactions between the Greek macroeconomic factors selected for this study, aiming at an in-depth analysis of the consequences of the Greek economy from the first Memorandum. The data were further processed using the Pearson correlation coefficient test, revealing a total of 20 statistically significant correlations, with 14 of them showing strong correlation characteristics at the statistically significant level of 0.01. These results highlight the dynamic properties of the model of behavior in times of crisis with the characteristics of the Greek economy.

**3. Results**

*3.1. Linear Regression Models*

Throughout the regression analysis, results regarding Greek macroeconomic impact, provoked by the 2010 Memorandum measures, will be deducted. To start with, at Table 1,

the most important descriptive statistics for the dependent variables used in analyzing the macroeconomic figures of the Greek economy are presented. For this purpose, the authors utilize the statistic metrics of min, max, the mean, and standard deviation for the equilibrium of current transactions, public debt, Government Budget, and GDPR. In Table 2, the Pearson correlation' coefficients are presented for all the variables used in the analysis, after running the propriate tests (Nettleton 2014). It should be noted that, prior to the development of the regression models, the necessary normality and collinearity tests were developed, according to Shapiro and Wilk's (Shapiro and Wilk 1965) process.

**Table 1.** Descriptive Metrics.

| Variables | Min | Max | Mean | St. Deviation |
| --- | --- | --- | --- | --- |
| Equil. of Current Transactions | −36,566 | −1318 | −14,000.56 | 11,815.63 |
| Public Debt | 162,937 | 442,613 | 366,527.74 | 100,185.82 |
| Government Budget | −35,981 | 2099 | −12,836.56 | 10,603.17 |
| GDPR | 93,064 | 241,990 | 175,306.4 | 39,804.9 |

**Table 2.** Correlation Analysis Matrix.

| | GDPR | Government Budget | Public Debt | Equil. of Current Transactions | Population | Net National Income | Equilibrium of Goods | Equil. of Primary Income | Antidamping Duties | Net Lending/Borrowing | Gross Labor Income | Foreign Exchange Reserves |
| --- | --- | --- | --- | --- | --- | --- | --- | --- | --- | --- | --- | --- |
| GDPR | 1 | −0.733 ** | −0.195 | −0.916 ** | 0.877 ** | 0.997 ** | −0.874 ** | −0.914 ** | 0.428 * | −0.679 ** | 0.962 ** | −0.670 ** |
| Government Budget | −0.733 ** | 1 | −0.030 | 0.597 ** | −0.878 ** | −0.690 ** | 0.442 | 0.613 ** | −0.062 | 0.515 * | −0.761 ** | 0.539 * |
| Public Debt | −0.195 | −0.030 | 1 | 0.439 | −0.227 | −0.297 | 0.482 * | 0.158 | 0.825 ** | 0.507 * | −0.418 | 0.714 ** |
| Equil. of Current Transactions | −0.916 ** | 0.597 ** | 0.439 | 1 | −0.754 ** | −0.936 ** | 0.943 ** | 0.880 ** | 0.653 ** | 0.988 ** | −0.959 ** | 0.624 ** |
| Population | 0.877 ** | −0.878 ** | −0.227 | −0.754 ** | 1 | 0.868 ** | −0.588 * | −0.651 ** | 0.224 | −0.694 ** | 0.918 ** | −0.601 ** |
| Net National Income | 0.997 ** | −0.690 ** | −0.297 | −0.936 ** | 0.868 ** | 1 | −0.917 ** | −0.886 ** | 0.393 | −0.699 ** | 0.968 ** | −0.699 ** |
| Equilibrium of Goods | −0.874 ** | 0.442 | 0.482 * | 0.943 ** | −0.588 * | −0.917 ** | 1 | 0.791 ** | 0.585 * | 0.927 ** | −0.910 ** | 0.687 ** |
| Equil. of Primary Income | −0.914 ** | 0.613 ** | 0.158 | 0.880 ** | −0.651 ** | −0.886 ** | 0.791 ** | 1 | 0.347 | 0.840 ** | −0.839 ** | 0.508 * |
| Antidamping Duties | 0.428 * | −0.062 | 0.825 ** | 0.653 ** | 0.224 | 0.393 | 0.585 * | 0.347 | 1 | 0.586 * | 0.205 | 0.419 |
| Net Lending/Borrowing | −0.679 ** | 0.515 * | 0.507 * | 0.988 ** | −0.694 ** | −0.699 ** | 0.927 ** | 0.840 ** | 0.586 * | 1 | −0.817 ** | 0.586 * |
| Gross Labor Income | 0.962 ** | −0.761 ** | −0.418 | −0.959 ** | 0.918 ** | 0.968 ** | −0.910 ** | −0.839 ** | 0.205 | −0.817 ** | 1 | −0.767 ** |
| Foreign Exchange Reserves | −0.670 ** | 0.539 * | 0.714 ** | 0.624 ** | −0.601 ** | −0.699 ** | 0.687 ** | 0.508 * | 0.419 | 0.586 * | −0.767 ** | 1 |

\* and ** indicate statistical significance at the 95% and 99% level.

Next, we move to the deployment of the first linear regression analysis, for the Greek equilibrium of current transactions. The regression of Greece's public debt with the main economic features is overall verified with a significant $p$-value below 0.01. This regression is expressed at Table 3 below, with $p$-value = 0.024 and $R^2$ = 0.852. Greek public debt variates up to 2.275, 3.195, and 1.536 from equilibrium of goods, net national income, and net lending/borrowing accordingly, since all these independent variables have significant $p$-values < 0.01. This means that, for every 1% increase of equilibrium of goods, net national income and net lending/borrowing, public debt increases by 227.5%, 319.5%, and 153.6%, respectively. At this point, based on public debts regression outputs, the authors can verify our first research Hypothesis, where the authors assume that public debt is significantly impacted by Greek economic features, affected by the 2010 Memorandum's measures.

**Table 3.** Greek economical factors' impact on public debt.

| Variables | Standardized Coefficient | $R^2$ | $t$-Test | F | $p$-Value |
|---|---|---|---|---|---|
| Constant | - | | 2.688 | | 0.024 * |
| Equilibrium of Goods | 2.275 | | 4.428 | | 0.003 ** |
| Net National Income | 3.195 | 0.852 | 4.581 | 8.268 | 0.003 ** |
| Net Lending/Borrowing | 1.536 | | 2.875 | | 0.024 * |

\* and \*\* indicate statistical significance at the 95% and 99% levels, respectively.

Above model's linear regression is shown below:

$$\text{Pub.Dept}_t = 2.275\text{Equil.of Goods}_t + 3.195\text{NetNat.Income}_t + 1.536\text{NetLend.Borrow}_t + e_t$$

In Table 4, the authors also distinguish the regression of Greece's current transactions equilibrium with the main economic features. It can be observed that the overall regression model and its independent variables are also generally confirmed with $p$-values equal to 0.000, below 0.01, and $R^2 = 0.998$. The fluctuation of Greek equilibrium of current transactions is 0.899 and $-0.134$ from net lending/borrowing and population, Greek equilibrium of current transactions increases by 89.9% and decreases by 13.4%, respectively. The above outcomes confirm the paper's second research Hypothesis, which indicates that the equilibrium of current transactions is significantly affected by the variation of main Greek economic features due to the 2010 Memorandum.

**Table 4.** Greek economical factors' impact on Equilibrium of Current Transactions.

| Variables | Standardized Coefficient | $R^2$ | $t$-Test | F | $p$-Value |
|---|---|---|---|---|---|
| Constant | - | | 6.185 | | 0.000 ** |
| Net Lending/Borrowing | 0.899 | 0.998 | 42.584 | 40.389 | 0.000 ** |
| Population | $-0.134$ | | $-6.355$ | | 0.000 ** |

\*\* indicate statistical significance at the 95% and 99% levels, respectively.

Model's linear regression is depicted as:

$$\text{Equil.CurrentTransactions}_t = 0.899\text{NetLend.Borrow}_t - 0.134\text{Population}_t + e_t$$

Moving to Table 5, again the authors see that the produced regression of Greek Government Budget (budget) is statistically significant in general, as well as every independent variable it contains, with $p$-values below 0.05 and $R^2 = 0.928$. The variation of Greek Government Budget (budget) from population and foreign exchange reserves is $-0.549$ and 0.464 respectively. Provided that population and foreign exchange reserves increase by 1%, Greek Government Budget will decrease by 54.9% and increase by 46.4% accordingly. With the verification of the third research hypothesis, assuming the significance of main Greek economic features, influenced by the 2010 Memorandum's measures, the effect on Greece's Government Budget is prominent.

**Table 5.** Greek economical factors' impact on Government Budget.

| Variables | Standardized Coefficient | $R^2$ | $t$-Test | F | $p$-Value |
|---|---|---|---|---|---|
| Constant | - | | 2.941 | | 0.000 ** |
| Population | $-0.549$ | 0.928 | $-3.415$ | 8.349 | 0.009 ** |
| Foreign Exchange Reserves | 0.464 | | 2.890 | | 0.020 * |

\* and \*\* indicate statistical significance at the 95% and 99% levels, respectively.

Government Budget's linear regression is:

$$\text{Govern.Budget}_t = 3.415\text{Population}_t + 2.890\text{For.ExchangeReserves}_t + e_t$$

In Table 6, the authors discern the results of GDPR's regression model with important features of the Greek economy. GDPR's regression comes as significant in general, due to *p*-value lower than 0.05 and $R^2$ = 0.998. Independent variables *p*-values are also lower than the significance level of 0.05, meaning they can explain the whole variation of GDPR. GDPR fluctuates up to 0.702, −0.132, 0.111, −0.024, and 0.087 from net national income, equilibrium of primary income, population, antidamping duties and gross labor income, respectively. Due to a potential increase in net national income, equilibrium of primary income, population, antidamping duties, and gross labor income, Greek GDPR increases by 70.2%, decreases by 13.2%, increases by 11.1%, decreases by 2.4%, and increases by 8.7% accordingly. Thus, we can confirm our fourth and last research hypothesis, according to which Greek GDPR is significantly impacted by the affected from 2010 Memorandum, Greece's main economic features.

**Table 6.** Greek economical factors' impact on GDPR.

| Variables | Standardized Coefficient | $R^2$ | *t*-Test | F | *p*-Value |
|---|---|---|---|---|---|
| Constant | - | | −7.400 | | 0.028 * |
| Net National Income | 0.702 | | 31.665 | | 0.000 ** |
| Equil. of Primary Income | −0.132 | 0.998 | −21.419 | 9.446 | 0.000 ** |
| Population | 0.111 | | 9.535 | | 0.000 ** |
| Antidamping Duties | −0.024 | | −4.325 | | 0.008 ** |
| Gross Labor Income | 0.087 | | 3.073 | | 0.028 * |

* and ** indicate statistical significance at the 95% and 99% levels respectively.

GDPR's produced linear regression model:

$$GDPR_t = 0.702NetNat.Income_t - 0.132Equil.ofPrim.Income_t + 0.111Population_t - 0.024Antidamp.Duties_t + 0.087GrossLabourIncome_t + e_t$$

### 3.2. Diagnostic and Exploratory Model

Fuzzy Cognitive Maps (FCMs) are fuzzy graph structures that are used to depict causality. Their ambiguity allows for ambiguous degrees of causation between ambiguous causal elements (Kosko 1986). It is a "soft computing" approach for modeling systems that blends fuzzy logic and neural networks. Even though the FCM generation approach is easily customizable, it is significantly reliant on human knowledge and know-how (Papageorgiou et al. 2003). FCMs are a parametric pattern of interpretation in which static features representing knowledge can be formed by clarifying basic framework attributes such as process variables, positive or negative correlations among variables, as well as the extent of connection which one factor may have to the other. The architecture of idea maps serve as the foundation for FCM research methodologies, which are carried out utilizing charts and diagram analyses between the elements in a system. Such frameworks may be used to simulate a process that is impacted by a large number of variables, with efforts made to map the correlation coefficients as well as the overall framework (Sharif and Irani 2006).

From the previous stages of analysis, data related to variables of the Greek economy were extracted, which significantly affected the course of its most important figures (GDP, Current Trading Balance, Public Debt, Budget). The data obtained refer to 28 independent variables, which will be tested for the importance of the influence and explanation of the course of the above-selected test dependent variables of the analysis to conclude the effectiveness of the measures of the first Memorandum. These data were further processed through the Pearson correlation coefficient test, with a total of 14 statistically significant correlations. Strong correlation characteristics are presented at a level of statistical significance of 0.01. These results underline the dynamic characteristics of the four most important explanatory variables of the Greek economy with several common independent variables. Through the variation of the statistical analysis variables, a causal association was found between the variables with high Pearson correlations (r > 0.8). The findings of the research using web analysis show strong correlations between the relevant variables of the Greek

economy and the dependent variables of GDP, Current Trade Balance, Public Debt, and Budget (Figure 4).

**Figure 4.** FCM of Greece's macroeconomic variables. Source: www.dev.mentalmodeler.com (accessed on 15 October 2021).

The direction of the arrow indicates the relationship between the variables and the width of the arrow is related to the gravity of the correlation. Positive correlations are shown with blue arrows and negative correlations with orange arrows. The above concept map was created using the MentalModeler (2022) software. The vague cognitive mapping of this dynamic environment provides better evaluation and explanatory opportunities for our study.

*3.3. FCM Simulation Model*

Having completed and illustrated the map of vague cognitive representation (Figure 4), the authors proceed to the creation and analysis of two scenarios through FCM to yield and represent the reaction of the basic quantities of the Greek economy. The FCM sigmoid function was chosen for these two scenarios, with similar methods being utilized in other studies (Sakas et al. 2022). Before the execution of the state forecast scenarios, the minimum and maximum levels for the correlation of the variables were set. For all variables in the sample of the Greek state of the economy, the value of 1 was set as the maximum and the value of −1 as the minimum. Given the nature of the variables, the negative values decreased in size from their previous levels. Therefore, the following two scenarios were performed as follows: the Pre-Crisis scenario (years 2006 to 2009) and the scenario following the imposition of Memorandum measures 2010 (years 2010 to 2012), based on the sample of individual variables collected during the years 2006 to 2012.

3.3.1. Pre-Crisis Scenario (2006–2009)

In the examined pre-crisis scenario of 2009, the selected quantities of the analysis of both the Greek economy and some other indicators show the following behavior, as described below. From the selected indicators increase, during the previous years of the crisis until its arrival (2006–2009), increase or improvement showed net national income, antidamping duties and gross labor income, foreign exchange reserves, and net lending/borrowing, with population showing a slight increase. On the other hand, the equilibrium of goods and the equilibrium of primary income decreased in the mentioned period.

For the two scenarios analyzed, proportional percentages of increase and decrease of the independent variables were used, regarding the figure of their change during the years 2006–2009 and 2010–2012. These percentages range between −0.4 to 0.4 depending on the degree of influence and importance of each independent variable in the dependent. For the period 2006–2009, gross labor income increased by 4%, antidamping duties by 3%, net lending/borrowing by 18%, net national income by 4%, foreign exchange reserves by 20%, the equilibrium of goods and the equilibrium of primary income increased by 5% and 30% accordingly, with population showing a small increase of 0.27%.

In the first scenario, the authors see that for the prices of the independent variables in the table in Figure 5, the dependent variables present the picture of the Greek economy in the period 2006–2009. More specifically, GDP and the public debt showed an increase of 2% and 3%, respectively, the government budget and the current transactions equilibrium decreased by 1% and 5%, respectively, with the variable of the General Economic situation of Greece having an increase of 1%. These results depict the situation of the main macroeconomic figures of Greece's economy through 2006–2009.

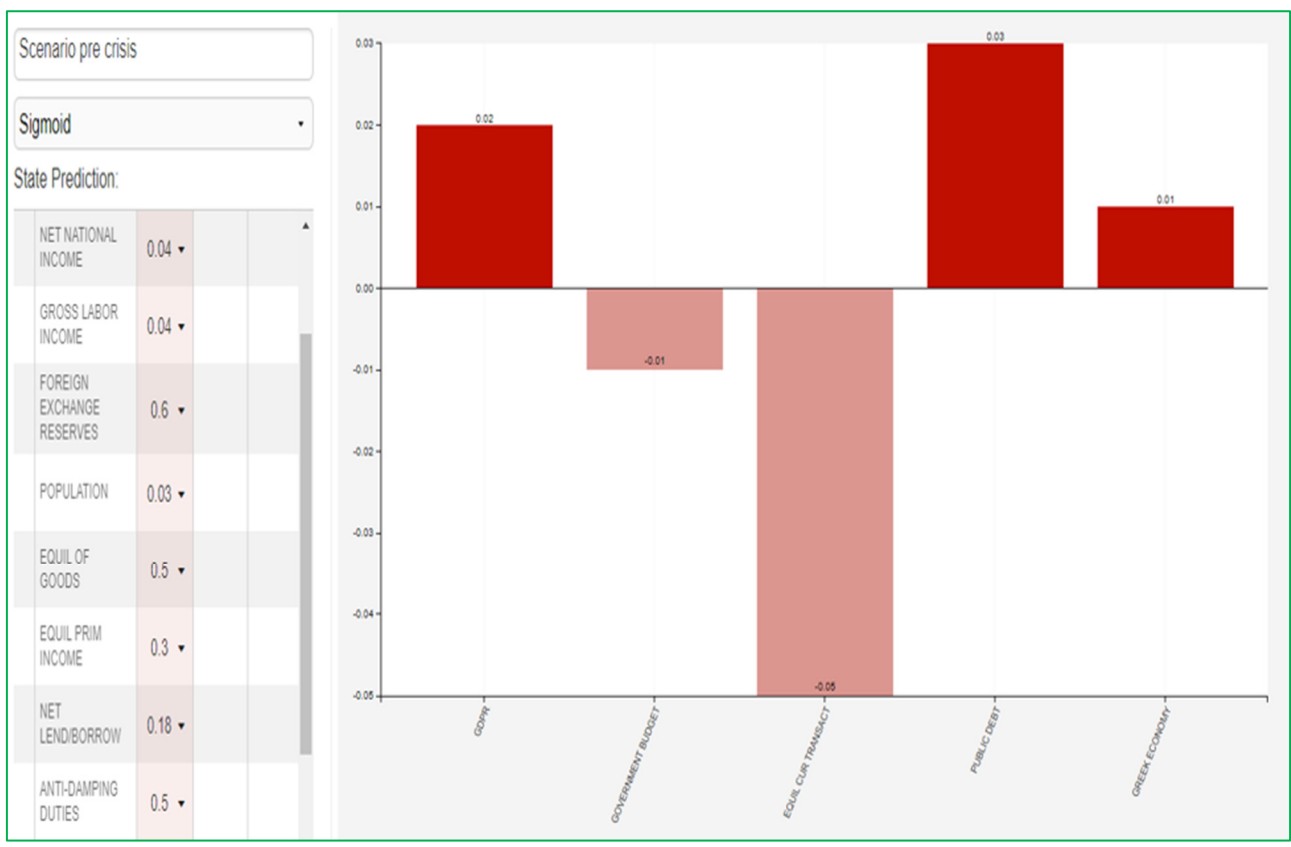

**Figure 5.** Illustration of the scenario of the economic situation of Greece before the crisis of 2009. Source: www.dev.mentalmodeler.com (accessed on 15 October 2021).

### 3.3.2. After Memorandum Measures Imposition Scenario (2010–2012)

The second scenario, which examines the period following the imposition of measures of the first Memorandum, i.e., between 2010 and 2012, first analyzes the changes in the size of the economy of that period, which significantly affect the dependent variables of the analysis (GDP, public debt, current transactions equilibrium, and budget). Antidamping duties and foreign exchange reserves increased in the period under review. Gross labor income, population, net lending/borrowing, net national income, equilibrium of goods, and primary income underwent a significant decrease.

Thus, for the period 2010–2012, the following percentages of change are attributed to the independent variables, depending on the fluctuation of each: 9% reduction of gross labor income, 0.2% reduction of population, net lending/borrowing decreased by 43%, 7% was the decrease net national income and equilibrium of goods and primary income decreased by 15% and 38% accordingly, with antidamping duties and foreign exchange reserves maintaining an increase of 6% and 13%, respectively.

Figure 6 shows the results of the predictive diagnostic model, with the table on the left containing the data analyzed above. The following is the result: Greece's GDP decreased by 12%, public debt increased but to a very small extent compared to previous years (3%), while both the government budget and the current transactions equilibrium increased by 11% and 6%, respectively. The variable of the Greek Economy that gives a more general assessment of the situation of the Greek market shows a decrease of 3%, again depicting the Greek economy's situation of the period 2010–2012.

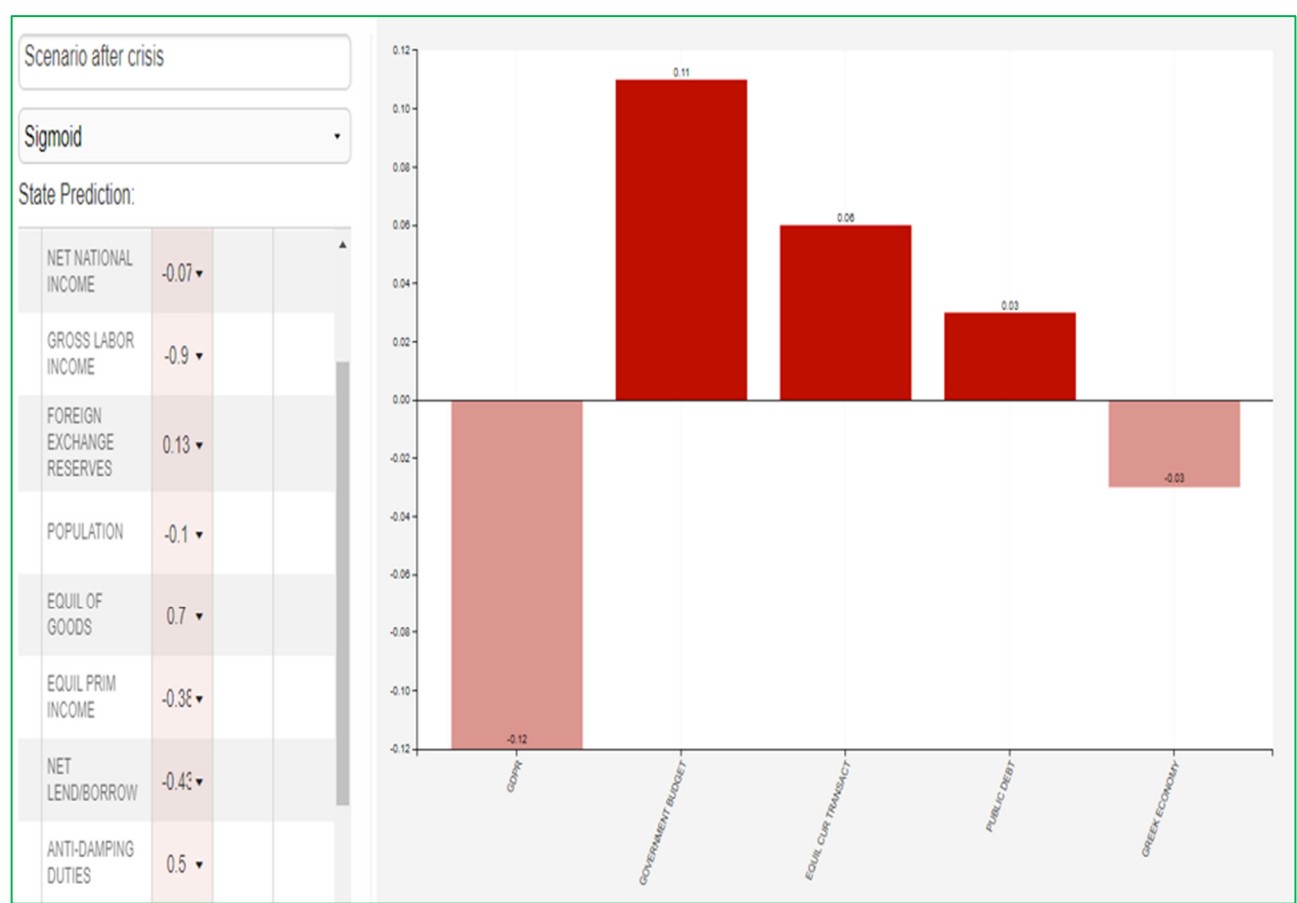

**Figure 6.** Illustration of the scenario of the economic situation of Greece after the first Memorandum 2010. Source: www.dev.mentalmodeler.com (accessed on 15 October 2021).

## 4. Discussion

The main results deducted from the data analysis section of the paper mostly focus on the relationships between various economic features and Greece's main macroeconomic figures. Firstly, from the linear regression and correlation analysis, it can be discerned that a specific group of Greek economic features (independent variables) are highly connected and express a significant amount of Greek GDPR, public debt, equilibrium of current transactions, and Government Budget (dependent variables). The produced regressions indicated that $R^2$ and model adjustment statistics are perfect (Tables 3–6), meaning that independent variables explain the total variance of the dependent ones (Freedman 2009). The group of independent features of the Greek economy mostly explaining its macroeconomic course contain the variables below: product taxes, imports of goods, balance of primary incomes, inflation, taxes on imports, central government expenditure, and net investment position of Greece, with antidamping duties and taxes.

As far as the FCM simulation model is concerned, results emerged showing the capability of the above economic features to fully explain the variation of the four selected key macroeconomic figures of Greece's economy. Before the outbreak of the financial crisis, the course of each feature is accurately calculated, giving a precise depiction of the course of GDPR, public debt, Government Budget, and equilibrium of current transactions. The overall economic situation of Greece, as described from these four figures, is slightly positive, presenting economic growth. The scenario run after the crisis and the Memorandum of 2010 again shows the capability of the selected economic features to express key Greek macroeconomic figures' variation and course.

The main outcome of the FCM simulation analysis is that even though figures like Government Budget and equilibrium of current transactions were highly improved, GDPR and public debt took a turn for the worst, deteriorating Greece's economic condition, as seen by the Greek Economy variable (sum of the four macroeconomic figures) in Figure 6. From the elaboration of the study, 8 main features of the Greek economy have been discerned that are capable of explaining and estimating the course of the 4 analyzed macroeconomic figures, including gross labor income, population, net lending/borrowing, net national income, equilibrium of goods, primary income equilibrium, antidamping duties, and foreign exchange reserves. Furthermore, the findings of the research strengthen the scope of FCM application as a simulation tool. By verifying the hypotheses set with the significant regressions performed, and the results of the simulation process, it is deducted that FCM simulation can provide a clear view of its utility and importance in the field of macroeconomics.

## 5. Conclusions

In this section, the results of the previous econometric analysis will be analyzed, with the ultimate goal of safely concluding the effects and results of the first Memorandum between Greece and the EZ. Through the analysis of the results, an attempt is made to provide an original methodology of explanation and conclusion regarding the effects of the first Financial Program on the basic economic features of Greece and the forecasting factors that explain these changes. Thus, using linear regression methods and FCM analysis, the effects and results of the 2010 Fiscal Program were initially estimated for the four figures of the Greek economy that showed the most significant wounds and represent the largest part of the performance of the Greek economy.

On this basis, one distinguishing trait of EZ nations hit by crises includes substantial and expanding current balance deficits throughout the years before the crises (Honkapohja 2014). The dominant assumption at that stage of the EZ's formation was that current balance deficits among member nations would not be a big issue in the monetary union (Blanchard and Giavazzi 2002). The financial view in the EZ, according to which Greek government debt was a secure investment, partly due to its prospect of the bailout by important nations, dampened the influence of government credit ratings of Greek cost

of interest. Simultaneously, reduced interest rates oiled the gears of financial growth, delivering the signal that there was no cost to accumulating government debt.

Each decrease in consumption had a greater impact on domestic items than on importing. As a result, the decrease in consumption for local goods had a greater impact on output than if the economy were much more dynamic. As a corollary, Greece seems to have a higher fiscal multiplier compared to other nations' crises with more open economies than the Greek one (Blanchard and Leigh 2013). The program's two key elements were budgetary restructuring and fundamental changes. Development has been delayed and unsuccessful in both of these sectors. Initially, the period of a monetary union is heavily dependent on the establishment of an error correction process; furthermore, engagement to a tough bond would not be a magic solution and cannot be sustained in the absence of the assistance of credible financial organizations (Eichengreen 1996).

Throughout this research, the authors found that most of the Greek economy features were affected negatively by the 2010 Memorandum measures, thus hurting key macroeconomic figures like GDPR, public debt, etc. Moreover, through simulation analysis, we have shown that by variating specific features of the Greek economy, a precise variation to the desired macroeconomic figures can occur. Based on macroeconomic figures (GDPR, public debt, etc.), features (e.g., net national income, equilibrium of primary income, population, antidamping duties, gross labor income, etc.), causal connection, the simulation results, and their course through the crisis and Memorandum years, it can be estimated that the 2010 Memorandum measures contributed to a further deterioration of Greek economy, separate from 2009 crisis effects. The referred 8 economic features (gross labor income, population, net lending/borrowing, net national income, equilibrium of goods, primary income equilibrium, antidamping duties, and foreign exchange reserves) have been found capable of satisfactorily predicating Greece's GDPR, public debt, equilibrium of current transactions, and Government Budget. This can be utilized in future research regarding Greek economy's reaction to upcoming global economic occurrences. Finally, the importance of Fuzzy Cognitive Mappings as a tool for macroeconomic projection and simulation should be highlighted, leading to a broader use of the method based on its efficiency.

## 6. Future Research

The ability to assess and estimate the effects of various economic features and figures by utilizing statistical and simulation models can prove valuable. Having analyzed the impacts of the 2010 Memorandum's measures on the Greek economy, through evaluation of vital economic figures' course, some key points for future research processing could be the addition of more simulation and prediction models for results validations. Furthermore, supplemental research should be carried out assessing the effects of the 2009 financial crisis in EU countries, and more specifically, discerning the differentiation of the crisis effects on countries of Southern Europe versus countries of Northern Europe.

**Author Contributions:** Conceptualization, S.P.M. and N.T.G.; methodology, D.P.S. and N.T.G.; software, S.P.M. and N.T.G.; validation, D.P.S. and G.K.; formal analysis, S.P.M.; investigation, N.T.G.; resources, S.P.M.; data curation, N.T.G.; writing—original draft preparation, S.P.M. and N.T.G.; writing—review and editing, N.T.G. and A.M.; visualization, N.T.G.; supervision, D.P.S. and G.K.; project administration, S.P.M. and N.T.G.; funding acquisition, S.P.M. All authors have read and agreed to the published version of the manuscript.

**Funding:** This research received no external funding.

**Institutional Review Board Statement:** Not applicable.

**Informed Consent Statement:** Not applicable.

**Data Availability Statement:** Data available in a publicly accessible repository. The data presented in this study are openly available in [www.statistics.gr and www.etc.europa.eu/eurostat] (accessed on 15 October 2021).

**Conflicts of Interest:** The authors declare no conflict of interest.

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
