# Peer review of "Analyzing Greece 2010 Memorandum’s Impact on Macroeconomic and Financial Figures through FCM"

_economies, doi:10.3390/economies10080178_

Round 1

Reviewer 1 Report

The topic is interesting but the authors' contribution is quite limited. First, there is no proper literature section to motivate the authors' contribution. Then, the authors' empirical analysis is based on univariate and linear regressions, this part is simplistic from an econometric and macroeconomic perspective. Then the authors use Fuzzy Cognitive Maps, but there is no innovation and their approach is not convincing. Finally, empirical evidences are poor. 

Author Response

On behalf of the authors, I would like to thank you for your reviews in our effort to publish our research and support our Ph.D. studies. We tried to adjust the paper according to your suggestions and provide you with sufficient justification for our methods and contribution. Please feel free to ask for any further clarifications. We also hope our comments satisfy your reports.

Reviewer 1

Responses

The topic is interesting but the authors' contribution is quite limited. First, there is no proper literature section to motivate the authors' contribution. Then, the authors' empirical analysis is based on univariate and linear regressions, this part is simplistic from an econometric and macroeconomic perspective. Then the authors use Fuzzy Cognitive Maps, but there is no innovation and their approach is not convincing. Finally, empirical evidences are poor. 

The authors’ contribution is determined by the collection of relative literature and data, the curation and validation of selected metrics, the performance of statistical tests and analyses, validation of performed tests and results, development of a Fuzzy Cognitive Mapping context followed by 2 scenarios that conclude to the paper’s main outcomes. Since our aim is to estimate the effects of 2010 MoU’s measures to Greek economy and FCM provides this result with immense effect and with data regression coefficients and correlations, paper’s goal is achieved.

A proper literature section has been added, highlighting the path that the authors followed to collect and analyze data in order to provide sufficient motivation for further elaboration (Lines 234 to 251).

The authors firstly deployed descriptive statistics, correlation and linear regression analysis, with the necessary statistical tests for examining data relationships, and due to data’s high grade of linear and causal relationships combined with the simplicity of the needed insights for the simulation analysis, did not perform any further nonlinear analyses.

As for the FCM analysis’ part, this tool is highly used and accepted for data simulation in various sectors (Lines 286 to 295), and that sparked the reason for selecting it. The innovation of FCM process applied to the research does not lie in its core application features, but in its implication to depict macroeconomic figures and provide a prediction for a specific time period, given the inputs of other macroeconomic variables. Furthermore, the authors added to the 2 scenarios’ figures the panel with the variables that were utilized and their values were adjusted in order to produce the simulation for the 4 figures of interest. This was added to improve the analysis’ results and provide the whole picture of the scenarios’ examined situations.

Regarding the empirical evidences, a rearrangement of the conclusions and discussions sections has been done and more focus on FCM’s role in the process has been given, apart from the assessment of the 2010 MoU measures’ effect on Greek economy, as can be seen in Lines from 542 to 550 and from 580 to 597. Empirical evidences do not focus on extensive analysis of these measures on the core activities of Greece’s economy, but only on the results of 4 main macroeconomic figures. To do so, the FCM  analysis and its multilevel connection ability, estimated the course of these 4 figures, which suits the observed situation of the economy. The authors did not luxuriate into further analysis that was never part of this study.

Reviewer 2 Report

I would add a bit more about the methodology and statistical sources used.

I would also add Graphs' sources (also in case they are authors' elaboration).

Author Response

On behalf of the authors, I would like to thank you for your reviews in our effort to publish our research and support our Ph.D. studies. We tried to adjust the paper according to your suggestions and provide you with sufficient justification for our methods and contribution. Please feel free to ask for any further clarifications. We also hope our comments satisfy your reports.

Reviewer 2

Responses

I would add a bit more about the methodology and statistical sources used.

I would also add Graphs' sources (also in case they are authors' elaboration).

A more extensive methodological approach has been added above the research hypotheses in Materials and Methods’ section, that focuses more on the methods and motivation that followed the research. In addition, information regarding the statistical sources used was also added at the Materials and Methods section, as well as the results section, according to reviewer’s suggestions, so as to further support the paper’s approach.

Regarding Graphs sources, more details were added to the footer of the Figure 1. Figure is authors’ elaboration and Figures 3 to 5 are both authors’ elaboration by utilizing the MentalModeler online application. In all cases, sources of Graphs were added.

Reviewer 3 Report

The authors study an interesting topic –  how Greece 2010 Memorandum affects macroeconomic and financial figures through FCM. They built four hypotheses and found that the Memorandum has been harsh and posed a negative effect on key Greek macroeconomic figures like GDPR, public debt, etc., especially with the ongoing 2008 financial crisis. My suggestions are as follows:

1. The regression eq. [1] – [4] are wrong: (1) there should have a time subscript; (2) there should have an error term at least given the data is time series – “the authors make effective use of the data collected from 252 the sample collected from the websites of ELSTAT [36] and EUROSTAT [37] and refers to 253 the years 2000 up to 2020”.

2. The method used in the paper is confusing. The authors say they used linear regression but they reported ANOVA in the results section. Linear regression and ANOVA are different methods: linear regression is used to estimate the change in the dependent variable given the change in independent variables, while ANOVA is used to find the difference in mean between variables of different groups.

3. On p. 7, the authors say the R-square is 1. However, in table 3, the R-square is only 0.998. The authors should report what the model shows.

Good luck with this interesting project!

Author Response

On behalf of the authors, I would like to thank you for your reviews in our effort to publish our research and support our Ph.D. studies. We tried to adjust the paper according to your suggestions and provide you with sufficient justification for our methods and contribution. Please feel free to ask for any further clarifications. We also hope our comments satisfy your reports.

Reviewer 3

Responses

The regression eq. [1] – [4] are wrong: (1) there should have a time subscript; (2) there should have an error term at least given the data is time series – “the authors make effective use of the data collected from 252 the sample collected from the websites of ELSTAT [36] and EUROSTAT [37] and refers to 253 the years 2000 up to 2020”.

Both error terms and time subscripts have been added to all regressions shown in the paper, according to reviewer’s suggestion.

The method used in the paper is confusing. The authors say they used linear regression but they reported ANOVA in the results section. Linear regression and ANOVA are different methods: linear regression is used to estimate the change in the dependent variable given the change in independent variables, while ANOVA is used to find the difference in mean between variables of different groups.

ANOVA tests were not relative to these regression models, so they were removed according to reviewer’s suggestions. These ANOVA models should have been more analytically referred, but due to major revision process, they were chosen not to be referred anymore.

On p. 7, the authors say the R-square is 1. However, in table 3, the R-square is only 0.998. The authors should report what the model shows.

Corrections were made according to reviewer’s comments. The regression’s R2 was wrongfully referred in the main body of the paper so it was modified accordingly.

Round 2

Reviewer 1 Report

Thanks for your appropriate answers.

Reviewer 3 Report

I very much appreciate the authors' revision efforts. The quality of the manuscript has been improved in this version. I recommend accepting this paper for publication.